# How Robust Is a Multi-Model Ensemble Mean of Conceptual Hydrological Models to Climate Change?

**Takayuki Kimizuka** [1,*] and **Yohei Sawada** [2]

1 Department of Civil Engineering, School of Engineering, The University of Tokyo, 7-3-1, Hongo Bunkyo-ku, Tokyo 113-8656, Japan

2 Institute of Engineering Innovation, School of Engineering, The University of Tokyo, 2-11-16 Yayoi, Bunkyo-ku, Tokyo 113-8656, Japan

* Correspondence: kimitaka1028@g.ecc.u-tokyo.ac.jp

**Abstract:** It is a grand challenge to realize robust rainfall-runoff prediction for a changing climate through conceptual hydrological models. Although multi-model ensemble (MME) is considered useful in improving the robustness of hydrological prediction, it has yet to be thoroughly evaluated. We evaluated the robustness of MME by 44 conceptual hydrological models in 582 river basins. We found that MME was more accurate and robust than each individual model alone. Although the performance of MME degrades in the validation period, the extent of degradation is smaller for MME than for individual models, especially when the climatology of river discharge in the validation period is greatly different from that in the calibration period. This implies the robustness of MME to climate change. It was found to be difficult to quantify the robustness of MME when the number of basins and models is small, which implies the importance of the large number of models and watersheds to evaluate the robustness and uncertainty in hydrological prediction.

**Keywords:** MARRMoT; rainfall-runoff analysis; robustness; multi-model ensemble; conceptual hydrological models; climate change



## 1. Introduction

A conceptual hydrological model represents the relationship between meteorological forcing such as precipitation and river discharge. It has been used for various administrative decisions, such as setting the height of levees. Hydrological models generally need to be calibrated to improve their accuracy by adjusting their parameters to reduce the difference between observed and simulated river discharge. However, when the climatic conditions of calibration periods are significantly different from the other periods, the performance of conceptual hydrological models may deteriorate due to over-learning in calibration periods (Duethmann et al., 2020 [1]; Zheng et al., 2018 [2]; Singh et al., 2011 [3]; Oudin et al., 2006 [4]; Fowler et al., 2016 [5]). As climate change progresses and meteorological forcing are expected to change in the future, conceptual hydrological models calibrated in the current climate may not be able to correctly project runoff in the future climate.

To overcome this issue, robust prediction methods, in which the prediction accuracy does not significantly degrade under a changing climate, have been intensively studied. Singh et al. (2011) [3] applied the concept of Prediction in Ungauged Basin (PUB) to improve the robustness of a conceptual hydrological model. They firstly calibrated model parameters in many river basins, and then interpolated parameters considering changes in climate forcing data when the projections of runoff in a future climate are implemented. Duethmann et al. (2020) [1] revealed that models which represent the evapotranspiration process more accurately by accounting for changes in vegetation have higher robustness to a changing climate. Deb and Kiem (2020) [6] showed that the distributed model is more robust than the lumped and semi-distributed models in a "differential split-sample test", in

which the robustness of models are evaluated using data from three different periods of wet, dry, and average years.

Multi-model ensembles (MME), in which several models with different structures are simultaneously used, have been shown to be effective. Duethmann et al. (2020) [1] thoroughly demonstrated that the uncertainty in the structure of models is the most important factor for the robustness of the projections of runoff. This result implies the effectiveness of MME in improving the robustness of runoff prediction since MME can consider the uncertainty in the structure of models. Therefore, in this study, we use MME, which is considered to be effective for robust runoff prediction. Zhang and Yang (2018) [7] performed MME prediction using eight rainfall-runoff models in the Yellow River basin and showed that the optimization method by a genetic algorithm can provide more reliable MME prediction in the validation period than individual models. Duan et al. (2006) [8] proposed the Bayesian model averaging method and showed that the weighing average of MME based on the likelihood of the models has high robustness. Shamseldin et al. (1996) [9] performed three methods to combine the output of five rainfall-runoff models, which are simple mean of MME, weighted mean of MME, and the nonlinear combination of MME by a neural network. They confirmed that the MME has higher robustness than individual models. However, in previous studies, the differences in forcing and river discharge data between the calibration and validation periods were not quantitatively discussed to validate the robustness of runoff projections. In order to investigate the robustness to climate change, it is necessary to evaluate the robustness of MME as a function of the differences of hydrometeorological conditions between calibration and validation periods. In addition, the previous studies used a small number of basins and models, so that the performance of MME was not thoroughly evaluated. Therefore, it is necessary to test the robustness of the MME projections to climate change using a large number of models in a large number of basins.

Recently, useful software toolboxes and datasets have been developed, which allow us to evaluate the MME predictions in a sufficiently large number of river basins with a sufficiently large number of models compared to previous studies. Knoben et al. (2019) [10] developed the Modular Assessment of Rainfall-Runoff Models Toolbox (MARRMoT), a toolbox capable of handling 46 models. Addor et al. (2017) [11] organized meteorological forcing and runoff data of 671 basins in the United States into the Catchment Attributes and Meteorology for Large-sample Studies Dataset (CAMELS dataset). Knoben et al. (2020) [12] used MARRMoT and the CAMELS dataset to evaluate the hydrological prediction performance of 36 models in 559 basins in the United States. They attributed the performance of individual models to geology and vegetation. In this paper, we will check whether the accuracy of MME is more robust than that of other individual models by applying 44 rainfall-runoff models to 582 river basins. We will also examine whether the superiority of MME over other individual models increases as the difference of hydrometeorological conditions between calibration and validation periods increases. Finally, the number of basins and models required to evaluate the performance of MME will be discussed with bootstrapping. It should be noted that this study does not propose MME prediction as the most accurate prediction method applicable for all regions; rather, this is a study to test the fundamental properties of MME prediction (e.g., robustness to climate change) compared to single-model predictions using a large number of models and basins.

## 2. Materials and Methods

### 2.1. Data and Model

The study areas were 582 basins selected from 671 basins in the continental United States contained in the CAMELS dataset (Figure 1). The CAMELS dataset was used to provide daily precipitation, temperature, potential evapotranspiration, and runoff from 1980 to 2014 for each basin, based on area-weighted calculations of observations. Since they have selected basins with homogeneous distribution of precipitation and minimal human disturbance, the target basins are inevitably small headwater basins. However, they are distributed across a wide

range of climatic zones in the U.S., from subtropical to frigid zones, and thus, the generality of the discussion in the following sections is not lost. There are three datasets: the Daymet dataset, with a spatial resolution of $1 \times 1$ km, and the NLDAS and the Maurer datasets, with a spatial resolution of $12 \times 12$ km. In this study, we used the Daymet dataset, which can consider the heterogeneity of forcing data in basins with complex topography.

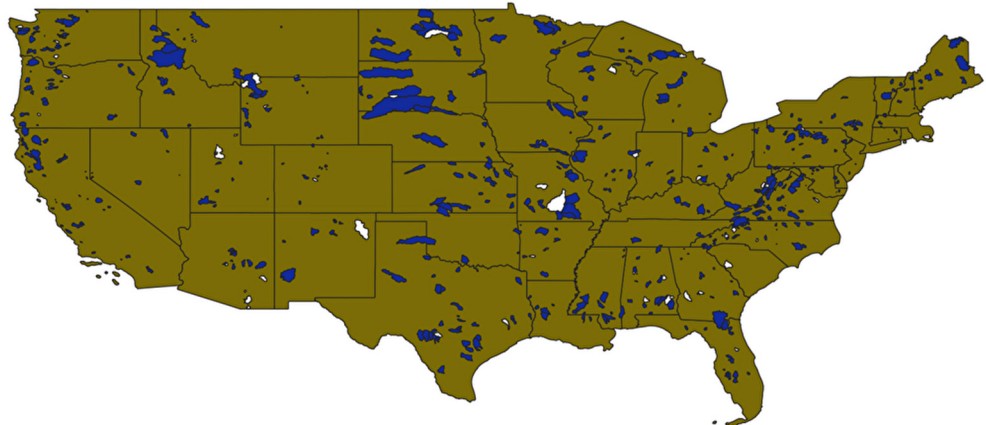

**Figure 1.** Tested basins in the CAMELS dataset. The blue basins were used, while the white basins were not.

The toolbox used to calculate runoff from meteorological data is the Modular Assessment of Rainfall-Runoff Models Toolbox (MARRMoT), which has 46 rainfall and runoff models available (Knoben et al., 2019) [10]. MARRMoT categorizes the numerous rainfall-runoff models used in previous studies into 46 models and organizes them in a way that makes them easy to use for testing model uncertainty. All models require precipitation, potential evapotranspiration, and temperature (only for models which have snow modules) as input variables, and generate runoff as an output variable. Since they are conceptual hydrological models, they should be applied to small basins. The parameters of each model have a set range and can be modified freely. In this study, the default parameters' ranges given by the software are used. The detailed information of individual models and their parameters can be found in the following MARRMoT supplement file: https://www.geosci-model-dev.net/12/2463/2019/gmd-12-2463-2019-supplement.pdf (accessed on 6 September 2022)

The reader can also refer to Table 1 and Figure 2 in Knoben et al. (2019) [10] for an overview of the models used. While the advantages and limitations of individual models can be seen there, it should be noted that MME is characterized by its statistical advantage over individual models, without looking at the behavior of individual models.

### 2.2. Experiments

Parameter calibration was conducted from 1 October 1980 to 28 September 1990 (3650 days), and the validation was conducted from 1 January 2013 to 31 December 2014 (730 days). We extracted and used 582 basins with complete data from 1 October 1980 to 31 December 2014. We set up the blank period between the calibration and validation periods since it is expected to have changes in hydrometeorological conditions from calibration periods to validation periods. In this study, the values of the parameters were not changed over time. Parameters in the conceptual hydrological models primarily depend on the geology and topography of river basins, so that they are theoretically unchanged over a short period of time (e.g., 100 years) due to climate change. However, land use and land cover changes might affect parameters, which were not considered in this work. In many cases, these anthropogenic changes are difficult to be predicted deterministically. Therefore, it is appropriate to consider the increase in uncertainty under a stationary hypothesis rather than accepting non-stationarity and changing the parameters, as mentioned

by Koutsoyiannis and Montanari (2014) [13]. No warm-up period for calibration was provided. The calibration method was the downhill simplex method. As explained in Lagarias et al. (1998) [14], for a model with $n$ parameters, generate an n-dimensional vector whose elements are the averages of the upper and lower limits of each parameter. Then, by adding 5%, 10%, ... and $5n$% to each element, $n$ vectors are created, and together with the first vector, $n + 1$ initial vectors are prepared. The objective function is then minimized by choosing one of the four transition methods (mirror image, expansion, contraction, and shrinkage). The estimation of parameters was iteratively updated by discarding the worst of the $n + 1$ estimated points. Although no range of estimated points should be specified in the usual downhill simplex method, in this study, we implemented the downhill simplex method with upper and lower bounds. If an element of the estimated vector was outside the upper or lower limit, the element was changed to the value of the upper or lower limit. The objective function is $-1 \times KGE$, where KGE (Kling–Gupta efficiency) is expressed by the following equation (Gupta et al., 2009) [15]:

$$KGE = 1 - \sqrt{(r-1)^2 + \left(\frac{\sigma_{sim}}{\sigma_{obs}} - 1\right)^2 + \left(\frac{\mu_{sim}}{\mu_{obs}} - 1\right)^2}, \tag{1}$$

where $r$ is the correlation coefficient between the observed and simulated values; $\sigma_{sim}$ and $\sigma_{obs}$ are the standard deviation of the simulated and observed values, respectively; and $\mu_{sim}$ and $\mu_{obs}$ are the mean values of the simulated and observed values, respectively. The maximum value of KGE is 1. For the convergence condition of the calibration, the tolerance of the estimation point and the tolerance of the objective function were both set to $5 \times 10^{-3}$. The stopping condition was set at 300 iterations and 600 function evaluations.

The MME prediction was generated by taking the ensemble average of the runoff predictions by the calibrated conceptual hydrological models in MARRMoT. We excluded model 40 (SMAR) and model 45 (PRMS) in our analysis because they were subject to instability in runoff calculations in multiple basins. We used 10 patterns of multi-model ensemble predictions, which were averages of the runoff predictions estimated by the models whose KGEs in the calibration period were more than 0.70, 0.65, 0,60, 0.55, 0.50, 0.40, 0.30, 0.20, 0.10, and 0. In addition, multi-model ensemble prediction of five patterns—each of which is an average of 10, 15, 20, 25, and 30 of the most accurate models in the calibration period—were used. When a selected model becomes unstable and its estimated runoff becomes divergent in the validation period, we excluded this model to calculate MME mean. We compared the prediction accuracy in the calibration and validation periods using KGE of 44 flow predictions by each individual model alone and the average of models narrowed down by KGE in the calibration periods. The estimated runoff is zero throughout the entire period in the limited case. In this case, KGE cannot be calculated because the correlation coefficient of the predictions cannot be obtained. Therefore, KGE was compared without the basins for which KGE could not be calculated. The MME evaluation procedure described so far is represented in Figure 2.

To examine the robustness of MME to climate change, the 582 basins were divided into three categories according to the difference of hydrometeorological conditions between the calibration and validation periods. We quantified the superiority of MME over each individual model in these three categories. As an indicator of the difference of hydrometeorological conditions, we used the absolute value of the difference of the mean value of the top 1% of daily runoff between the calibration and validation periods. We divided the absolute value of the difference by the mean of top 1% daily runoff in the calibration periods to calculate the index. Basins were classified into those with large rates of change (>0.4), moderate rates of change (0.2 to <0.4), and small rates of change (<0.2).

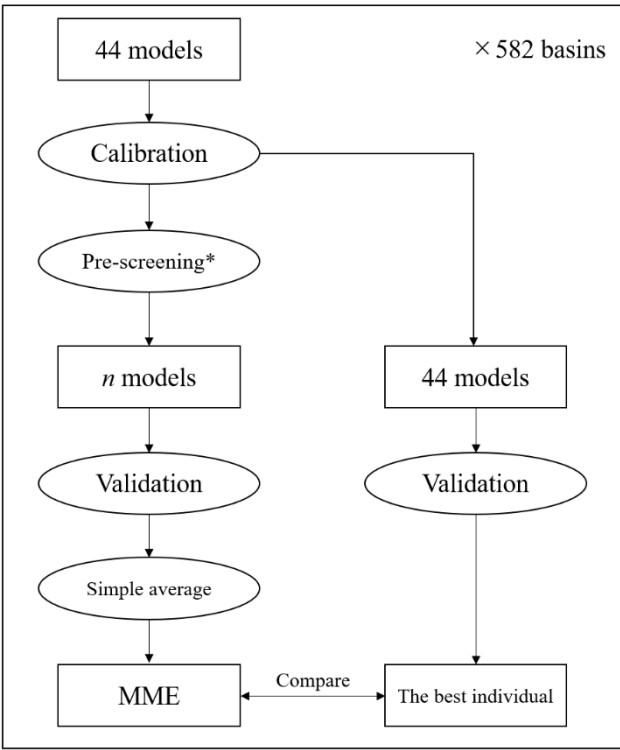

*Models are narrowed down based on KGE

| Threshold | 0, 0.1, 0.2, 0.3, 0.4, 0.5, 0.55, 0.6, 0.65, 0.7 |
|-----------|--------------------------------------------------|
| Rank      | top 30, top 25, top20, top15, top 10             |

**Figure 2.** Flowchart of MME evaluation.

### 2.3. Bootstrapping

Bootstrapping was conducted to discuss the number of basins and models required to verify the evaluation of the MME's performance. For the bootstrapping of the number of basins, 10, 50, 100, and 200 basins out of the 582 basins used in the study were randomly selected 1000 times. For the bootstrapping of the number of models, 5, 10, 20, and 30 models out of the 44 models used in the study were randomly selected 1000 times. The MME performance evaluation index $S$ was defined and calculated for each selected subset of basins and models.

$$S = \underset{j}{\mathrm{median}}\left(KGE_{MME_j}\right) - \underset{i}{\mathrm{max}}\left(\underset{j}{\mathrm{median}}\left(KGE_{model_{i,j}}\right)\right) \tag{2}$$

$KGE_{MME_j}$ is KGE in the validation period calculated using MME in basin $j$ (in the basin subset). For bootstrapping on the number of basins, MME of the top 10 models of KGE in the calibration period were averaged. For bootstrapping on the number of models, MME of the models with KGE greater than 0 in the calibration period was averaged due to the limited number of models available. $KGE_{model_{i,j}}$ is KGE in the validation period calculated using model $i$ (in the model subset) in basin $j$ (in the basin subset). The first term on the right-hand side of Equation (2) represents the accuracy of the MME, and the second term represents the accuracy of the best model. Therefore, a positive value of S means that the MME prediction is more accurate than the best individual model in the validation period, and a negative value of S means the opposite. In other words, the larger S is, the more accurate the MME is than each model alone.

## 3. Results

### 3.1. Calibration Results for Individual Models and MME

Figure 3 shows a comparison of KGE between each of the 44 individual models alone and 15 different MMEs with various thresholds. For one example of MME prediction, the spatial distribution of KGE in the validation period when the MME's threshold of KGE (see Section 2.2) is set to 0 is shown in Figure 4. Although models' performance can be evaluated by a variety of criteria, model 34 (FLEX-IS) is the best performing model when comparing medians of KGE in the validation period in 582 basins. This model consists of five tanks: a snow melt module, interception storage, unsaturated zone, and slow and fast runoff reservoir. Knoben et al. (2020) revealed that the model performs relatively well, especially in baseflow-dominated and snowy basins, but performs poorly in the other basins. Compared with 44 box plots of individual models' prediction, the MME by threshold 0.7 was the highest in the median, the first quartile, and the third quartile for both the calibration and validation periods. In the calibration period, the median, the first quartile, and the third quartile of KGE of MME with 0 KGE threshold were ranked as the 8th, 2nd, and 13th highest, respectively, in all 44 individual models. In the validation period, the median, the first quartile, and the third quartile of KGE of MME with 0 KGE threshold were ranked as the 3th, 2nd, and 6th highest compared with all 44 individual models, respectively, while those of model 34 (FLEX-IS) were ranked as the 1st, 3rd, 1st highest, respectively. The degradation of the runoff prediction skill of MME from calibration periods to validation periods is much less than that of individual models. In other words, the robustness of MME was confirmed considering the diversity of basins and models to the maximum extent possible.

When the threshold of models to be included in MME was lowered and more models with low accuracies in the calibration period were used, the accuracy in the validation period was slightly lower than that of some models comparing the median. However, the first quartile was sufficiently high compared to KGE of each model alone. Although MME tends to be inferior to the other accurate models because models with poor accuracy can be included in MME, as the number of models used increases, the predictions are less likely to be so bad that KGE becomes negative, and MME has a much lower risk of exhibiting low performance than the individual models. In the calibration period, KGE of MME becomes significantly higher when a few accurate models are included into MME. However, in the validation period, the number of models used to generate MME brings no significant difference in the prediction accuracy. In addition, the variability of KGE tended to be smaller for the MME by a small number of accurate models in the calibration period, while the difference of the variance of KGE between different MMEs was small in the validation period.

These findings have been verified by the Kolmogorov–Smirnov test, which tests whether the two sets of data are from different probabilistic distributions at the 5% significance level. Tables 1–4 show the results of the Kolmogorov–Smirnov test for the combinations of KGEs of two MMEs with different averaging methods depending on the threshold and rank. KGEs in the calibration period are judged to be different even if the thresholds and ranks of KGEs by which the models are selected to the MME are close to each other. In contrast, KGEs in the validation period are judged to be different populations only when the thresholds and ranks are substantially different. Results shown in Figure 3 and Tables 1–4 quantitatively indicate that the more strictly the models used for the MME are narrowed down to a small number of models with high accuracy in the calibration period, the more accurate the MMEs are in both calibration and validation periods. However, the results also imply that the accuracy in the validation period does not improve as much as expected from the improvement in the accuracy in the calibration period.

Tables 1–4, results of the Kolmogorov–Smirnov test to determine whether the distributions of KGEs of MME taken with different criteria belong to different population distributions. Circles indicate different distributions, and crosses indicate the same distribution.

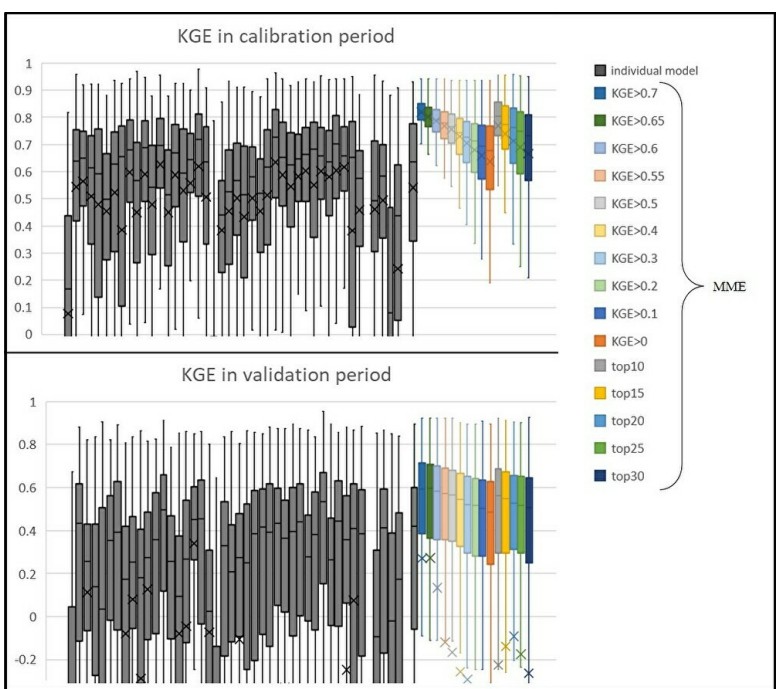

**Figure 3.** Box plots of KGEs predicted by each model alone (gray box) and by MME (colored box) for all 582 basins for which KGEs could be calculated. KGEs for the calibration period are shown in the top panel, and KGEs for the validation period are shown in the bottom panel. For MME, the 10 series on the left are MME which selected models by the threshold of KGE during the calibration period, and the 5 series on the right are MME which selected models by the rank of KGE during the calibration period.

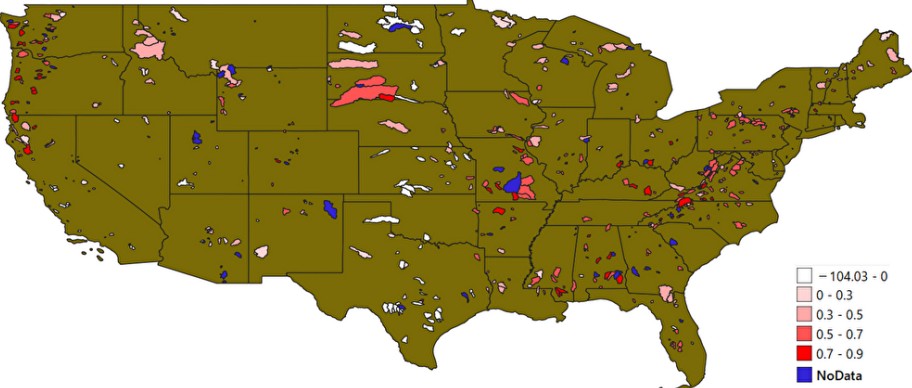

**Figure 4.** Distribution of KGE in the validation period of MME by the model with a threshold of 0. The darker the red, the higher the KGE and the more robust the basin. Blue basins are basins that do not have complete data for either the calibration period or the validation period.

**Table 1.** The result when MME is taken based on the different threshold of KGE. Also, it shows the differences for KGE in the calibration period.

| Threshold | 0.7 | 0.65 | 0.6 | 0.55 | 0.5 | 0.4 | 0.3 | 0.2 | 0.1 | 0 |
|---|---|---|---|---|---|---|---|---|---|---|
| 0.7 | × | ○ | ○ | ○ | ○ | ○ | ○ | ○ | ○ | ○ |
| 0.65 | ○ | × | ○ | ○ | ○ | ○ | ○ | ○ | ○ | ○ |
| 0.6 | ○ | ○ | × | ○ | ○ | ○ | ○ | ○ | ○ | ○ |
| 0.55 | ○ | ○ | ○ | × | × | ○ | ○ | ○ | ○ | ○ |
| 0.5 | ○ | ○ | ○ | × | × | ○ | ○ | ○ | ○ | ○ |

**Table 1.** *Cont.*

| Threshold | 0.7 | 0.65 | 0.6 | 0.55 | 0.5 | 0.4 | 0.3 | 0.2 | 0.1 | 0 |
|---|---|---|---|---|---|---|---|---|---|---|
| 0.4 | ○ | ○ | ○ | ○ | ○ | × | ○ | ○ | ○ | ○ |
| 0.3 | ○ | ○ | ○ | ○ | ○ | ○ | × | × | ○ | ○ |
| 0.2 | ○ | ○ | ○ | ○ | ○ | ○ | × | × | × | ○ |
| 0.1 | ○ | ○ | ○ | ○ | ○ | ○ | ○ | × | × | × |
| 0 | ○ | ○ | ○ | ○ | ○ | ○ | ○ | ○ | × | × |

**Table 2.** The result when MME is taken based on the different threshold of KGE. Also, it shows the differences for KGE in the validation period.

| Threshold | 0.7 | 0.65 | 0.6 | 0.55 | 0.5 | 0.4 | 0.3 | 0.2 | 0.1 | 0 |
|---|---|---|---|---|---|---|---|---|---|---|
| 0.7 | × | × | × | × | ○ | ○ | ○ | ○ | ○ | ○ |
| 0.65 | × | × | × | × | ○ | ○ | ○ | ○ | ○ | ○ |
| 0.6 | × | × | × | × | × | ○ | ○ | ○ | ○ | ○ |
| 0.55 | × | × | × | × | × | × | ○ | ○ | ○ | ○ |
| 0.5 | ○ | ○ | × | × | × | × | ○ | ○ | ○ | ○ |
| 0.4 | ○ | ○ | ○ | × | × | × | × | × | ○ | ○ |
| 0.3 | ○ | ○ | ○ | ○ | ○ | × | × | × | × | × |
| 0.2 | ○ | ○ | ○ | ○ | ○ | × | × | × | × | × |
| 0.1 | ○ | ○ | ○ | ○ | ○ | ○ | × | × | × | × |
| 0 | ○ | ○ | ○ | ○ | ○ | ○ | × | × | × | × |

**Table 3.** The result when MME is taken based on the different rank of KGE. Also, it shows the differences for KGE in the calibration period.

| Ranking | Top10 | Top15 | Top20 | Top25 | Top30 |
|---|---|---|---|---|---|
| top10 | × | ○ | ○ | ○ | ○ |
| top15 | ○ | × | ○ | ○ | ○ |
| top20 | ○ | ○ | × | × | ○ |
| top25 | ○ | ○ | × | × | × |
| top30 | ○ | ○ | ○ | × | × |

**Table 4.** The result when MME is taken based on the different rank of KGE. Also, it shows the differences for KGE in the validation period.

| Ranking | Top10 | Top15 | Top20 | Top25 | Top30 |
|---|---|---|---|---|---|
| top10 | × | × | ○ | ○ | ○ |
| top15 | × | × | × | × | ○ |
| top20 | ○ | × | × | × | × |
| top25 | ○ | × | × | × | × |
| top30 | ○ | ○ | × | × | × |

*3.2. Robustness to Change in Hydrometeorological Conditions*

Figure 5 summarizes the KGE of 582 river basins in the validation period, dividing them into three classes of basins (high, moderate, and low changes in hydrometeorological conditions) from the calibration period to the validation period. The classes with small, moderate, and large changes in the top 1% runoff include 223, 152, and 193 basins, respectively. The total number of basins is less than 582 because we excluded basins with missing runoff data for either the calibration period or the validation period. The MME shown in

the red boxplots is MME using the models with the top 10 KGE in the calibration period. Compared to the individual models, the MME has relatively high KGE in all categories. When comparing the median KGE of the 44 models and those of MME, the MME has the second highest KGE in basins with small changes in hydrometeorological conditions and the highest KGE in basins with moderate and large changes in hydrometeorological conditions. As the magnitude of changes in the top 1% runoff increases, KGE becomes smaller for all individual models and MME. For example, model 34 (FLEX-IS), which was the best model in the validation period, has a median KGE of −0.09 for basins with large hydrometeorological changes, which is significantly lower than for basins with smaller changes. However, the magnitude of KGE's decrease in MME is smaller than individual models. Most of the individual models show a large decrease in KGE with increasing changes in top 1% runoff, and the number of basins whose KGE was more than zero was just 47% of the river basins with large hydroclimatic changes when calculated with model 34 (FLEX-IS), while MME shows KGE greater than zero in 124 basins (over 60%). Although the predictions of individual models are reliable for basins with small and moderate changes in hydrometeorological conditions, for basins with large changes in hydrometeorological conditions, the reliability of the predictions by individual models substantially degrades. In this case, MME has a greater advantage.

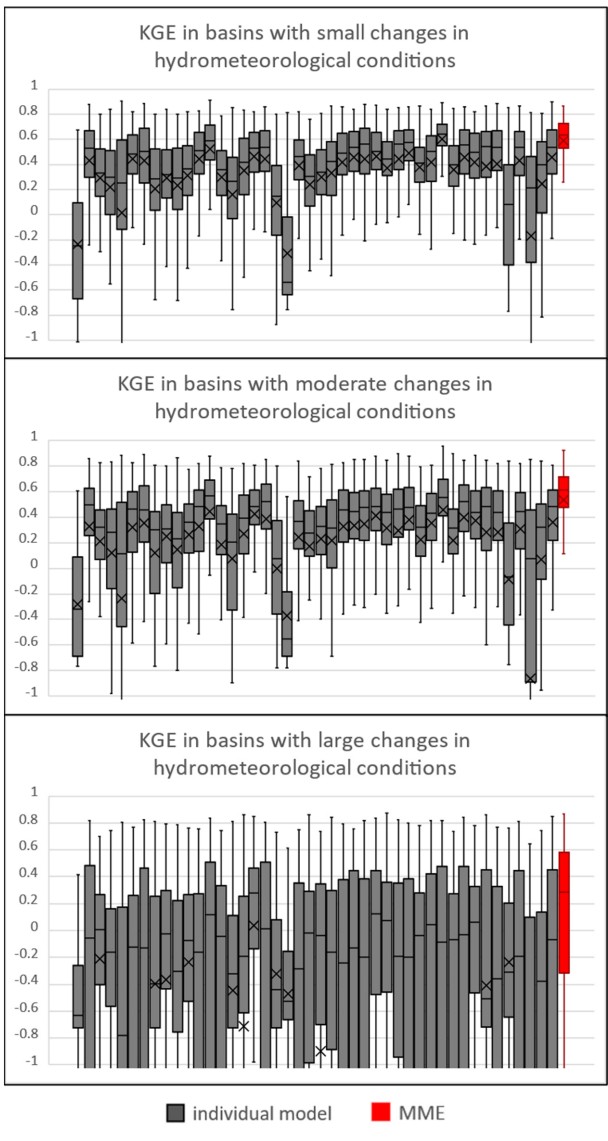

**Figure 5.** Box plots of KGE during the validation period in basins with small, moderate, and large

changes in top 1% runoff. The absolute value of the change rate in the average of the top 1% runoff from the calibration period to the validation period was used as an indicator of changes in hydrometeorological conditions. Basins with a value of less than 0.2 were considered to have small changes in hydrometeorological conditions, basins with a value between 0.2 and 0.4 were considered to have moderate changes in hydrometeorological conditions, and basins with a value greater than 0.4 were considered to have large changes in hydrometeorological conditions (see Section 2.2). MME (red box) is based on the top 10 models of KGE in the calibration period.

### 3.3. Robustness of the Robustness of MME's Skill

Figure 6 shows histograms of the MME performance evaluation index S described in Section 2.3. As we described in Section 2.3, we calculated S 1000 times by randomly selecting sub-samples of basins and models, to show the relative frequencies of S values. A positive value of S means that the MME predicts more accurately than the best individual model in the validation period. It should be noted that since it is generally impossible to know in advance which is the best model in the validation period, a negative S does not necessarily mean that the performance of the MME is inferior to the prediction of each model alone.

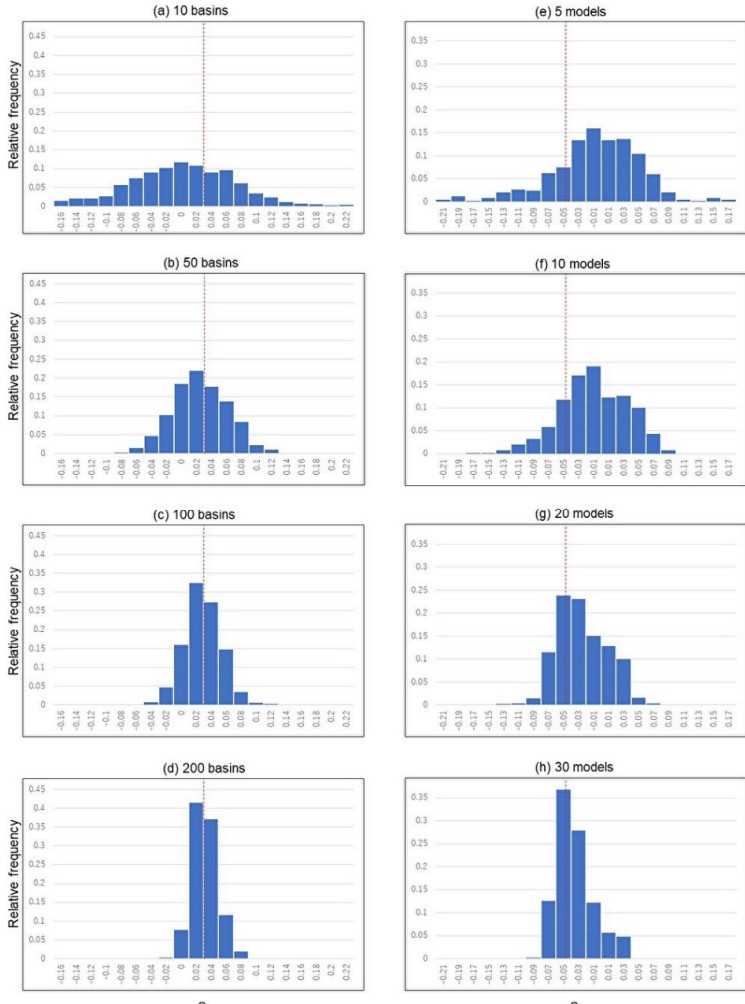

**Figure 6.** Histogram by bootstrapping of the number of basins and the number of models. The basins were randomly selected as (**a**) 10, (**b**) 50, (**c**) 100, and (**d**) 200 basins. The models were randomly selected as (**e**) 5, (**f**) 10, (**g**) 20, and (**h**) 30 models. Each random selection was repeated 1000 times, and the relative frequencies of the MME performance evaluation index S are shown. The red line shows the index S calculated with all basins and all models.

Figure 6a–d shows the results of bootstrapping of the number of basins. The peaks become higher and sharper as the number of basins used increases. As the number of basins is increased, the spread of the histogram becomes smaller, although the trend slows down as the number of basins is increased. In the case of 10 basins, the peak is at a different location from the other cases and the variance of estimated S is extremely large, indicating that the estimated performance of MME can be substantially biased when the number of basins used is small. For example, if the S-value is calculated using only 10 basins, the S-value can be negative depending on the basins used, resulting in an underestimation of the performance of the MME. Figure 6e–h shows the results of bootstrapping of the number of models. The higher the number of models is, the higher and sharper the peak is. When the number of models is small, the histogram becomes flat. As the number of models increases, the spread of the histogram becomes smaller, although the trend slows down as the number of models increases, which is the same as the bootstrapping of the number of basins. When the number of models is small, S tends to be overestimated. In other words, when more models are used, a smaller S-value tends to be obtained. This is because as the number of models increases, the relatively accurate individual model out of 44 models can be easily selected. The results shown in Figure 6 imply that if the MME is evaluated with a small number of basins and models (such as 10 basins and 5 models), the results may significantly differ from the results obtained with a larger number of basins and models.

## 4. Discussion

In this study, we used a much larger number of models to generate the MME prediction than previous studies, and we verified the robustness of MME to climate change by considering the uncertainty in the model structure. The performance of MME was evaluated using the data of a much larger number of river basins than previous studies. Knoben et al. (2020) [12] calibrated 36 models with 559 basins and examined the loss of accuracy over the validation period. Similarly, here, we thoroughly considered model- and basin-specific uncertainties and provided general results about the performance of MME, which are independent to the selection of models and basins.

Our results support many previous works with a relatively small number of models and basins. As has been shown by many previous works (e.g., Zhang and Yang (2018) [7], Duethmann et al. (2020) [1], Zheng et al. (2018) [2], Singh et al. (2011) [3], Oudin et al. (2006) [4]), the accuracy of runoff prediction by individual models and MME in the validation period is significantly lower than that in the calibration period. It was also shown that there was a large variation in the accuracy of predictions in both the calibration and validation periods due to differences in the structure of the model, as shown by Deuthmann et al. (2020) [1], who pointed out that imperfections in the model structure were the most significant cause of the decrease in accuracy. Guo et al. (2017) [16] also pointed out that the accuracy of the projections of runoff under climate change vary greatly depending on the structure of the models, such as evapotranspiration schemes.

In this study, we used many models, which makes it possible to select the models to be included to calculate the mean of MME. However, the number of models used to generate a good MME is unclear. In this issue, there are two contradictory points of view. First, the use of many models, which is one of the advantages of MME, cancels out the bias of individual models and improves the prediction performance. Second, the use of many models leads to the adoption of inaccurate models in MME, which reduces the prediction performance of MME. Our results indicated that no matter what threshold or ranking of the models used, they tended to produce a higher KGE in the validation period than the predictions made by each individual model. This result is important and useful because it shows that MME can achieve high robustness in the validation period regardless of the selection method and the criteria of the models.

In this study, MME prediction was calculated as a simple average of the selected models. However, there are other methods of calculating the average of MME, such as Bayesian model averaging (Duan et al., 2007) [8] and the weighted average and the

neural network method described by Shamseldin et al. (1996) [9]. Using not only linear but also nonlinear combining methods, a more robust MME prediction method than this study can be found. In addition, the robustness of MME prediction based on conceptual hydrological models evaluated in this study should be compared with the robustness of distributed hydrological models to deepen the understanding of the uncertainty in hydrological prediction.

## 5. Conclusions

In this study, we thoroughly evaluated the performance and robustness of MME of conceptual hydrological models using 44 individual models in 582 river basins. We found that MME is highly accurate and robust, and that its superiority to individual models does not depend on the thresholds to select models to be included in MME. In addition, while the accuracy of all individual models' prediction was low when changes in hydrometeorological conditions from the calibration period to the validation period were substantially large, MME could maintain stable and high accuracy in such cases, implying its robustness to climate change. Although MME is not always the best performing model in each river basin, we can conclude that MME is a robust choice for climate change impact assessment in which the best performing model cannot be identified in advance. The results of our bootstrapping experiment showed that it is necessary to perform experiments with a large number of models and basins in order to reach these conclusions.

**Author Contributions:** Conceptualization, T.K. and Y.S.; methodology, T.K. and Y.S.; software, T.K.; validation, T.K.; formal analysis, T.K.; investigation, T.K. and Y.S.; resources, Y.S.; data curation, T.K. and Y.S.; writing—original draft preparation, T.K.; writing—review and editing, Y.S.; visualization, T.K.; supervision, Y.S.; project administration, Y.S.; funding acquisition, Y.S. All authors have read and agreed to the published version of the manuscript.

**Funding:** This research was funded by JST FOREST program (grant no. JPMJFR205Q) and the Foundation of River and basin Integrated CommunicationS (FRICS).

**Institutional Review Board Statement:** Not applicable.

**Informed Consent Statement:** Not applicable.

**Data Availability Statement:** Modular Assessment of Rainfall-Runoff Models Toolbox (MARRMoT) v3 (Knoben et al., 2019): https://zenodo.org/record/3552961#.YaT9MdDMI2w (accessed on 6 September 2022). Catchment Attributes and Meteorology for Large-sample Studies Dataset (CAMELS DATASET) (Newman et al., 2014): https://ral.ucar.edu/solutions/products/camels (accessed on 6 September 2022).

**Conflicts of Interest:** The authors declare no conflict of interest.

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
