# Peer review of "How Robust Is a Multi-Model Ensemble Mean of Conceptual Hydrological Models to Climate Change?"

_water, doi:10.3390/w14182852_

Round 1

Reviewer 1 Report

Abstract:

The following sentence is not clear: “This superiority 14 of MME becomes more apparent as the climatology of river discharge in the validation period is 15 more different from that in the calibration period.” What do you mean with “apparent”?. Authors should better clarify this sentence. The abstract should be auto explicative, and then a reader would prefer to well understand in this part all the highlights, without analysing successive parts of the manuscript….

 Sect. 2.2

Lines 125-128: the sentence is not very clear. Please clarify

 WHOLE MANUSCRIPT

It is not very clear how is modelled the change of parameters along the time. By using “step functions” for each parameter? Is it checked if stationary hypothesis (with specific values of parametric sets) can be suitable in a changing climate?

Authors should better clarify these aspects!

 Concerning the possibility of adoption of stationary hypothesis for climate changes context, I suggest these references:

 Koutsoyiannis, D., Montanari, A.: Negligent killing of scientific concepts: the stationarity case. Hydrol. Sci. J. 60, 1174–1183 (2014)

De Luca, D.L.; Petroselli, A.; Galasso, L. (2020). Modelling climate changes with stationary models: is it possible or is it a paradox? In: Sergeyev Y., Kvasov D. (eds) Numerical Computations: Theory and Algorithms. NUMTA 2019. Lecture Notes in Computer Science, vol 11974. Springer, Cham. https://link.springer.com/chapter/10.1007/978-3-030-40616-5_7

Serinaldi, F., Kilsby, C.G.: Stationarity is undead: uncertainty dominates the distribution of extremes. Adv. Water Resour. 77, 17–36 (2015)

Reviewer 2 Report

Though the topic is interesting, the paper has come out without any head and tail!.

1) Authors talk about MME, but not mentioning which are those models?.

2) No detailed explanations on the methodological framework - there should be a flow chart/ block diagram showing the details methodology.

3) When MME is mentioned, authors should bring out the most significant and best performing models, its advantages and limitations.

4) Otherwise, simply mentioning MME, what is the meaning without giving any details.

5) The results are for very small basins.

6) The results should be verified according to various characteristics of basins.

The paper needs much improvement and clarity before further consideration.

Round 2

Reviewer 2 Report

Though the revised paper is improved, it has not considered all previous comments.

1) Provide an Appendix with the models considered in the study.

2) Atleast provide a comparison with the best performing model and its variation with MME results.

3) Do necessary changes in the conclusions and mention about the best performing model and MME variation and show why MME is better.
